# LLM-Exp: Exploring the Policy in Reinforcement Learning with Large Language Models

## Abstract

Policy exploration is critical in training reinforcement learning (RL) agents, where existing approaches include the $\epsilon$-greedy method in deep Q-learning, the Gaussian process in DDPG, etc. However, all these approaches are designed based on prefixed stochastic processes and are indiscriminately applied in all kinds of RL tasks without considering any environment-specific features that influence the policy exploration. Moreover, during the training process, the evolution of such stochastic process is rigid, which typically only incorporates a decay of the variance. This makes the policy exploration unable to adjust flexibly according to the agent's real-time learning status, limiting the performance. Inspired by the analyzing and reasoning capability of LLM that reaches success in a wide range of domains, we design **LLM-Exp**, which improves policy exploration in RL training with large language models (LLMs). During the RL training in a given environment, we sample a recent action-reward trajectory of the agent and prompt the LLM to analyze the agent's current policy learning status and then generate a probability distribution for future policy exploration. We update the probability distribution periodically and derive a stochastic process that is specialized for the particular environment, which can be dynamically adjusted to adapt to the learning process. Our approach is a simple plug-in design, which is compatible with DQN and any of its variants or improvements. Through extensive experiments on the Atari benchmark, we demonstrate the capability of LLM-Exp to enhance the performance of RL. Our code is open-source at `https://anonymous.4open.science/r/LLM-Exp-4658` for reproducibility.

## 1 Introduction

In recent decades, reinforcement learning (RL) has achieved unprecedented development and is proven to be a powerful tool for training smart agents in solving sequential decision-making problems (Sutton, 2018; François-Lavet et al., 2018). The success of deep RL is especially noteworthy in tasks with high complexity, such as game playing (Silver et al., 2017; Vinyals et al., 2019; Berner et al., 2019; Ye et al., 2021), chip design (Mirhoseini et al., 2021), smart city governance (Hao et al., 2021; 2022; 2023; Zheng et al., 2023; Wang et al., 2024b), and mathematical reasoning (Fawzi et al., 2022), where deep RL agents now exhibit performance surpassing human professionals in more and more scenarios. In the training of RL agents, policy exploration plays an indispensable role, which allows the agents to sample a diverse range of actions and uncover better strategies that may not be immediately apparent. The explore-exploit trade-off is a critical aspect of reinforcement learning, where agents must balance exploring new possibilities to improve long-term rewards and exploiting known strategies to maximize immediate gains.

Various policy exploration approaches have been proposed in existing RL algorithms, including $\epsilon$-greedy in DQN (Mnih et al., 2015), Gaussian process noise in DDPG (Lillicrap et al., 2016), and probability distribution sampling in PPO (Schulman et al., 2017). Despite their success, existing policy exploration methods have notable limitations. First, they are designed based on prefixed stochastic processes that are applied uniformly across all kinds of tasks without any environment-specific adaption, neglecting the unique characteristics of different environments that may influence policy exploration. Besides, the evolution of these stochastic processes during training tends to be simplistic, which typically merely involves a gradual decay in variance over time. As a result, these methods fail to flexibly adjust the policy exploration strategy based on the agent's real-time

learning status, potentially reducing the effectiveness of policy exploration, especially in complex or non-stationary environments.

There exist several major challenges in addressing these limitations. First of all, RL tasks span diverse environments, and the training process involves a vast number of action steps, during which the agent's learning status undergoes complex changes. Thus, relying on more fine-grained manual designs based on prefixed stochastic processes becomes increasingly impractical. Fortunately, the emergence of large language models (LLMs) (Zhao et al., 2023; Wu et al., 2023) provides an opportunity to overcome this challenge. Such LLMs are capable of automatically analyzing the agent's real-time learning status at a high frequency, enabling more dynamic and intelligent adjustments to policy exploration without the need for manual intervention. However, the majority of RL tasks involve environmental states as images, and the training process typically covers millions of frames. This presents the problem for common multimodal LLMs, which are often limited to processing a single image per prompt and are computationally expensive.

Facing these challenges, we propose to enhance the policy exploration in RL based on LLMs, namely **LLM-Exp**. In LLM-Exp, during the RL training process within a given environment, we periodically sample recent action-reward trajectories from the agent's experience and prompt the LLM to analyze the agent's current policy learning status based on the trajectories. The LLM then generates a tailored probability distribution that guides future policy exploration based on the agent's learning status and the specific characteristics of the environment. We update the probability distribution regularly, allowing it to dynamically adapt as the agent progresses through training and ensuring the exploration strategy evolves in response to changes in learning status. By doing so, we derive a specialized stochastic process from this dynamically updated distribution, which is uniquely suited to the environment, and we actually replace the prefixed ones used in traditional methods with it. In our approach, the LLMs operate entirely with textual inputs and outputs, reducing the computational overhead and making it compatible with various existing types of LLMs. Besides, our approach is designed to be a simple plug-in, which can be seamlessly integrated with DQN and any of its variants or improvements (Schaul et al., 2016; Van Hasselt et al., 2016; Wang et al., 2016; Fortunato et al., 2018; Hessel et al., 2018; Laskin et al., 2020) without the need for any significant architectural changes, making it a versatile solution for various RL tasks. We conduct extensive experiments on the Atari benchmark (Bellemare et al., 2013; Kaiser et al., 2019), where the results demonstrate the capability of LLM-Exp to enhance the performance of various RL algorithms.

In summary, the main contributions of this work include:

- We propose LLM-Exp, a method that leverages LLMs with purely textual inputs and outputs to dynamically adjust the policy exploration during RL training, which addresses the limitations of traditional policy exploration with prefixed stochastic processes.

- Our approach is designed as a simple plug-in, allowing seamless integration with DQN and any of its variants, enabling enhanced exploration without requiring significant modifications to existing RL architectures.

- We conduct extensive experiments to validate the effectiveness of our method, demonstrating its ability to improve the policy exploration across various RL tasks and environments.

## 2 PRELIMINARIES

### 2.1 MARKOV DECISION PROCESS (MDP)

Markov decision process (MDP) is the fundamental framework for reinforcement learning, where an agent solves the decision-making problems in interaction with a dynamic environment. Mathematically, an MDP is defined by a tuple $(\mathcal{S}, \rho, \mathcal{A}, P, R)$ with $\mathcal{S}$ representing the state space, and $\rho \in \Delta(\mathcal{S})$ denoting the probability distribution of initial state, where $\Delta(\mathcal{S})$ is a collection of probability distribution over $\mathcal{S}$. $\mathcal{A}$ denotes the action space, and when executing a specific action in a given state, $P : \mathcal{S} \times \mathcal{A} \rightarrow \Delta(\mathcal{S})$ and $R : \mathcal{S} \times \mathcal{A} \rightarrow \mathbb{R}$ are the state transition probability function and the single-step reward function, respectively. At time step $t$, the agent executes action $a_t \in \mathcal{A}$ under the state of $s_t \in \mathcal{S}$, and then receives a reward of $r_t$ and experiences the state transition to $s_{t+1}$. The agent's goal in an MDP is to maximize its cumulative reward over time, which is the sum of discounted single-step rewards. This cumulative reward at time step $t$ is formalized as

$G_t = \sum_{k=0}^{\infty} \gamma^k r_{t+k}$, where $\gamma$ is the discount factor that determines the importance of future rewards. To achieve this, the agent needs to balance exploiting known strategies and exploring unknown ones, where the former one means selecting the action with the largest estimated cumulative reward. In contrast, the latter one requires trying other possibilities with randomness.

## 2.2 DEEP Q-LEARNING

One of the most established methods for solving RL tasks is the Deep Q Networks algorithm (Mnih et al., 2015), which trains a neural network $Q_\theta$ to approximate the agent's action-reward mapping. DQN updates the parameters of $Q_\theta$ by minimizing the error between predicted reward from $Q_\theta$ and its greedily estimated target value:

$$\mathcal{L}_\theta^{DQN} = \left( Q_\theta(s_t, a_t) - \left( r_t + \gamma \max_{a'} Q_\theta\left(s_{t+1}, a'\right) \right) \right)^2. \tag{1}$$

Specifically in DQN, policy exploration is achieved by the $\epsilon$-greedy mechanism, where most of the time, the agent executes $a_t$ that maximizes $(Q_\theta(s_t, a_t)$, while with a small probability of $\epsilon$, the agent randomly selects $a_t$ from the action space.

Various improvements have been made to improve the original DQN. Prioritized experience replay (Schaul et al., 2016) improves data efficiency by adding importance sampling into the replaying buffer. Double-DQN (Van Hasselt et al., 2016) modifies the target value, namely $(r_t + \gamma \max_{a'} Q_\theta(s_{t+1}, a'))$, by substituting $Q_\theta$ with the target network $Q_{\theta'}$, which is a delayed copy of $Q_\theta$ to avoid overestimation. Dueling-DQN (Wang et al., 2016) improves the network structure of $Q_\theta$ to decouple the state value from the advantage of taking a given action in that state. Noisy-DQN (Fortunato et al., 2018) introduces noisy networks, which inject randomness directly into the network of $Q_\theta$, allowing for better policy exploration. Ultimately, Rainbow (Hessel et al., 2018) consolidates these improvements into a single combined algorithm, and CURL (Laskin et al., 2020) enhances the performance of Rainbow by adding an unsupervised contrastive learning target.

## 2.3 LARGE LANGUAGE MODELS (LLMs)

Large language models are sophisticated neural networks with billions of parameters, which are mainly trained by predicting the probability of the next word in a sequence. Given $\{w_1, w_2, ..., w_{t-1}\}$, the model output $w_t$ to maximize the observation likelihood in the corpus as:

$$\prod_{t=1}^{T} P(w_t | w_1, w_2, ..., w_{t-1}). \tag{2}$$

Over the past few years, LLMs have made significant progress, where notable examples include the GPT family (Brown et al., 2020; Kalyan, 2023; Achiam et al., 2023), the Llama family (Touvron et al., 2023; Dubey et al., 2024), the PaLM family Chowdhery et al. (2023), etc. These LLMs have exhibited strong capability across a wide range of natural language processing tasks, ranging from text generation and translation to summarization and question answering (Zhao et al., 2023; Chang et al., 2024).

# 3 METHODS

## 3.1 OVERVIEW

In this paper, we propose to improve the policy **Exp**loration in RL based on **LLM**s, namely **Exp-LLM**. As shown in Figure 1, our framework employs two LLMs that collaborate through natural language communication and guide the policy exploration through a structured process. First, we introduce the basic task description and sample action-reward trajectories of the agent from the previous episode, prompting the former LLM to summarize the learning status of the agent and recommend potential exploration strategies (Section 3.2). Then, we feed the obtained summary and suggestion to the second LLM, which subsequently generates a probability distribution for policy exploration in the next $K$ episodes (Section 3.3). Here, $K$ is the hyper-parameter representing the interval at which the probability distribution is updated. It is worth mentioning that in a substantial

number of RL tasks, such as the Atari benchmark, the environmental states are represented by RGB images, but in our design, we only sample the actions and rewards of the agent and exclude the states. Therefore, our LLMs only receive textual inputs, reducing the computational consumption and ensuring compatibility with either multi-modal or text-only LLMs.

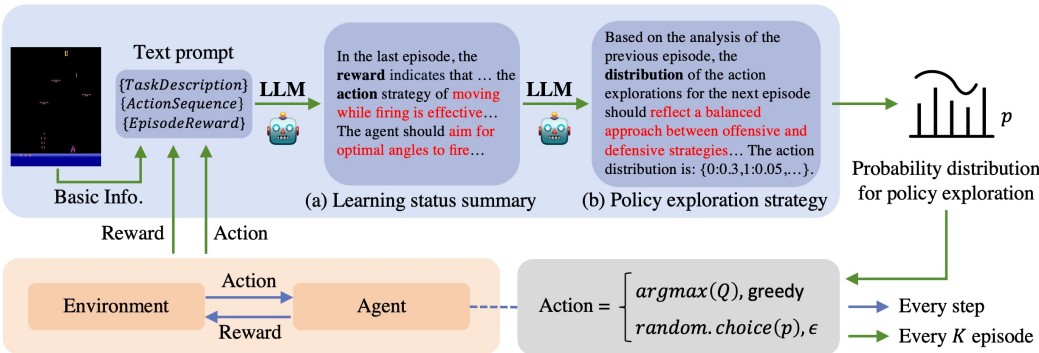

Figure 1: Illustration of our Exp-LLM method that enhances policy exploration in RL with LLMs.

## 3.2 LEARNING STATUS SUMMARIZING

To effectively guide the policy exploration, we design the first LLM to summarize the learning status of the agent every $K$ episode and provide suggestions on future exploration (Figure 1a). To achieve this, we first describe the basic elements of the task as {*TaskDescription*}, ensuring that outputs of the LLM align with the environmental characteristics.

**Task Description**: The task is a reinforcement learning problem where an agent {*TaskDetails*}. The action space is discrete with {*ActionDim*} options: {0: {*Action0*}, 1: {*Action1*}, ...}. {*ActionDetails*}. The observation space consists of raw pixel values representing the game screen, showing the {*ObservationElements*}. The agent receives a reward of {*RewardDetails*}. The game ends when {*EndConditions*}. The goal is to {*GoalDetails*}.

Then, at each time of updating, we sample $M$ actions uniformly from the latest episode, obtaining {*ActionSequence*}, where $M$ stands for the sampling density. We also extract the total reward of the latest episode, obtaining {*EpisodeReward*}. Combining these, we design a tailored prompt for the first LLM, as formulated below:

**Prompt 1**: You are describing the last episode of the training process on a task. {*TaskDescription*}. In the last episode, the total reward is {*EpisodeReward*}, and the action sequence extracted at intervals is {*ActionSequence*}. Please analyze the data, generate a description, and provide possible strategy recommendations.

This prompt provides the necessary context for the LLM to summarize the information in the previous episode and extract meaningful insights into the agent's learning status. Additionally, it requires the LLM to offer potential strategy recommendations, aiming at providing more useful information for the upcoming policy exploration strategy generation process.

## 3.3 POLICY EXPLORATION STRATEGY GENERATION

To improve policy exploration, we design the second LLM in our framework to generate a probability distribution over the action space for future exploration (Figure 1b). This distribution is generated based on the first LLM's analysis regarding the learning status of the agent in the previous episode, as well as its suggestions for future policy exploration. We feed this information into the second LLM through the prompt structured as follows:

**Prompt 2**: You are determining the probability distribution for action exploration in reinforcement learning. {*TaskDescription*}. Here is a description of the situation in the previous episode:

{*Summary&Suggestions*}. Based on the above information, please analyze what kind of actions should be selected to better improve the task effectiveness. Please output the distribution of the {*ActionDim*} action explorations for the next episode based on your analysis in decimal form. Your output format should be: {1: [probability], 2: [probability], ...}.

Based on this prompt, the LLM analyzes which actions should be selected and outputs a probability distribution for the next $K$ episode's policy exploration. This process enables the agent to prioritize actions that are more likely to improve the performance while also increasing the exploration of previously underexplored actions to discover new strategies. By periodically updating the strategy every $K$ episode, we ensure that the policy exploration evolves dynamically to adapt to the agent's learning progress.

## 4 EXPERIMENTS

### 4.1 EXPERIMENTAL SETTINGS

We evaluate the performance of LLM-Exp on Atari (Bellemare et al., 2013; Kaiser et al., 2019), a widely used benchmark for evaluating RL algorithms. In the main experiments, we use the Double-DQN algorithm (Van Hasselt et al., 2016) as the basis and insert our LLM-Exp into it. To verify the performance of LLM-Exp in enhancing the raw RL algorithm, we selected 15 environments from the 26 environments in Atari, where the training of the raw Double-DQN algorithm can converge stably and obtain good rewards. In addition, we set the number of training steps to 100k-500k across different environments based on how fast the reward increases when training the original Double-DQN algorithm. In our deployment, we fix a set of hyper-parameters across all environments. Specially, we use GPT-4o mini[1] as the core LLM and set the two key parameters in our design, namely action sampling density and exploration adjusting interval, as $M = 100$ and $K = 1$. For reproducibility, we provide specific values of all hyper-parameters in Appendix A.1 and list detailed contents of the prompts in Appendix A.3.

### 4.2 OVERALL PERFORMANCE

Table 1: Performance of LLM-Exp on the Atari benchmark, where the results are recorded at the end of training and averaged across 3 random seeds. The bold fonts indicate the best results.

| Environment | Double-DQN | | Double-DQN+LLM-Exp | | Improvement (%) |
|---|---|---|---|---|---|
| | Score | Human-norm score (%) | Score | Human-norm score (%) | |
| Alien | 245.46 | 0.26 | 268.44 | **0.59** | 126.92 |
| Amidar | 22.34 | 0.97 | 26.75 | **1.22** | 25.77 |
| BankHeist | 18.64 | 0.6 | 19.51 | **0.72** | 20.00 |
| Breakout | 2.67 | 3.36 | 2.74 | **3.62** | 7.74 |
| ChopperCommand | 840.63 | 0.45 | 868.33 | **0.87** | 93.33 |
| CrazyClimber | 17070.76 | 25.11 | 17694.35 | **27.6** | 9.92 |
| Freeway | 5.25 | 17.75 | 20.64 | **69.71** | 292.73 |
| Hero | 1439.7 | 1.38 | 2689.62 | **5.58** | 304.35 |
| Jamesbond | 60.84 | 11.63 | 77.35 | **17.66** | 51.85 |
| Krull | 2933.05 | 125.06 | 3009.12 | **132.19** | 5.70 |
| MsPacman | 411.07 | 1.56 | 489.9 | **2.75** | 76.28 |
| Pong | -15.71 | 14.13 | -14.13 | **18.61** | 31.71 |
| Qbert | 306.07 | **1.07** | 301.97 | 1.04 | -2.80 |
| Seaquest | 201.58 | **3.18** | 196.15 | 3.05 | -4.09 |
| UpNDown | 1370.99 | 7.51 | 1489.54 | **8.57** | 14.11 |
| Total-Mean | 1660.89 | 14.27 | 1809.35 | **19.59** | 37.27 |
| Total-Median | 245.46 | 3.18 | 268.44 | **3.62** | 13.84 |

---

[1]https://platform.openai.com/docs/models/gpt-4o-mini

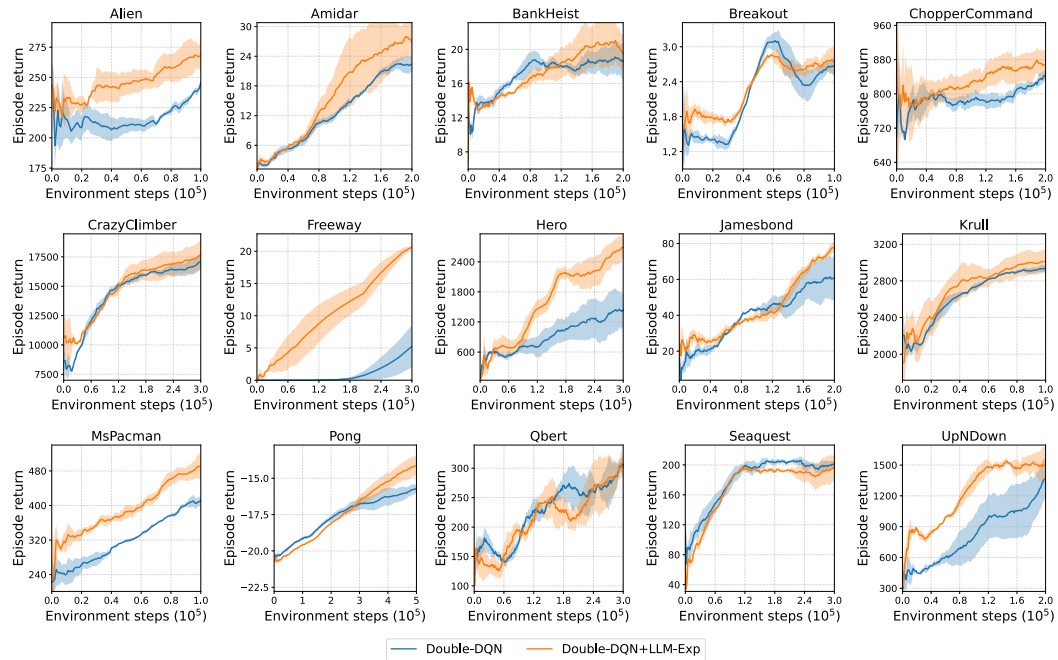

Figure 2: Performance of LLM-Exp on the Atari benchmark. In each experiment, we repeatedly run the training process with three different random seeds and use the shaded area to indicate the standard deviations.

We train agents using the Double-DQN algorithm and Double-DQN + LLM-Exp in the aforementioned environments, where in each environment, we repeat the training process with three different random seeds and average the results. We show the learning curves for each environment in Figure 2 and summarize the game scores obtained at the end of training in Table 1. To better compare the games with varying score ranges and difficulty levels, we also normalize the game scores using the average score of human players (Cagatan & Akgun, 2024; Yarats et al., 2021). The results indicate that LLM-Exp improves the human-normalized score in 13 out of 15 environments, with an increment of 37.27% and 13.84%, respectively, on the mean and median score, verifying its ability to enhance the performance of the existing RL algorithm.

## 4.3 COMPATIBILITY WITH DIFFERENT RL ALGORITHMS

Table 2: Compatibility of LLM-Exp with various RL algorithms. The human-norm scores (%) are recorded at the end of training and averaged across 3 random seeds. The underlined results indicate improvements over the raw RL algorithm, and the bold fonts indicate the best results.

| Environment | DQN | | PER-DQN | | Dueling-DQN | | Rainbow | | CURL | |
|---|---|---|---|---|---|---|---|---|---|---|
| | Raw | LLM-Exp | Raw | LLM-Exp | Raw | LLM-Exp | Raw | LLM-Exp | Raw | LLM-Exp |
| Alien | 1.00 | 1.11 | 0.51 | 0.55 | 0.39 | 0.61 | 0.24 | 0.70 | 3.21 | **3.62** |
| Freeway | 22.41 | 71.64 | 12.02 | 66.42 | 24.18 | 60.38 | 38.05 | 47.33 | 75.19 | **78.9** |
| MsPacman | 2.06 | 2.48 | 2.12 | 2.44 | 1.39 | 2.09 | 1.48 | 1.6 | **6.08** | 5.99 |

In our design, LLM-Exp is a simple plug-in method which can be seamlessly integrated with DQN and any of its variants or improvements. To verify, besides the Double-DQN algorithm aforementioned, we selected another five widely applied variants or improvements of DQN, including the vanilla DQN (Mnih et al., 2015), DQN with prioritized experience replay (PER-DQN) (Schaul et al., 2016), Dueling-DQN (Wang et al., 2016), Rainbow (Hessel et al., 2018), and CURL (Laskin et al., 2020). From the above environments, we selected three environments with relatively good training outcomes as representatives, namely the environments of Alien, Freeway, and MsPacman. In the

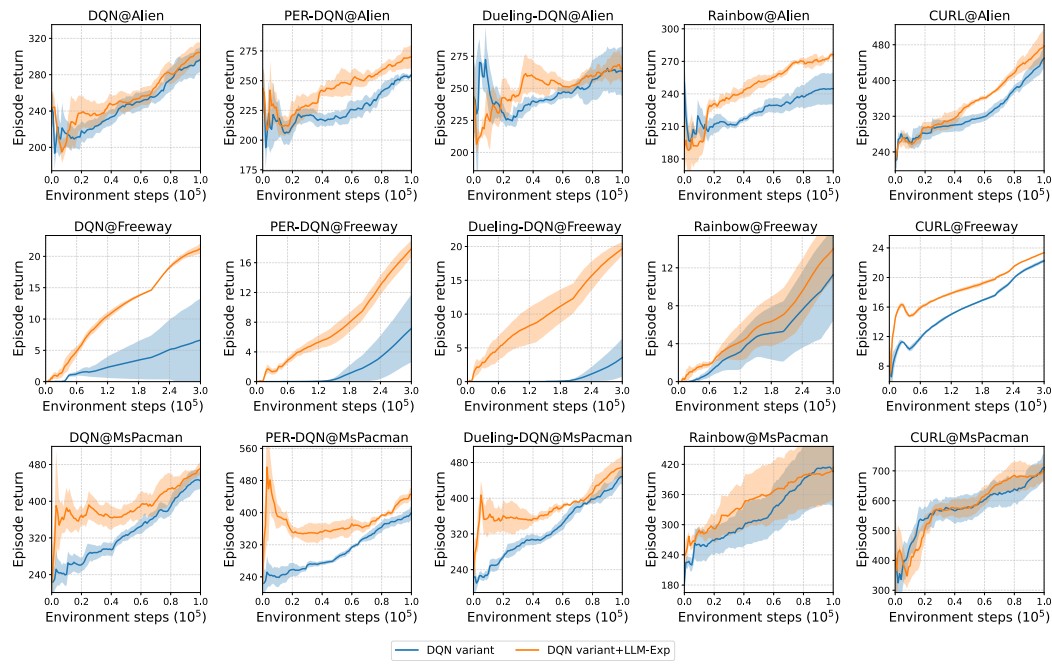

Figure 3: Compatibility of LLM-Exp with various RL algorithms. In each experiment, we repeatedly run the training process with three different random seeds and use the shaded area to indicate the standard deviations.

three environments, we train agents with the original versions of the five RL algorithms, as well as the versions integrating our LLM-Exp method with each of them. In each experiment, we repeat the training process with three different random seeds and average the results. We show the learning curves for the 15 experiments (5 algorithms×3 environments) in Figure 3 and summarize the game scores obtained at the end of training in Table 2. As the results illustrate, different variants or improvements of DQN exhibit diverse performance in different environments, while LLM-Exp consistently improves the human-normalized score of the original algorithms (14 out of 15 experiments). This proves LLM-Exp's compatibility with various RL algorithms, indicating its potential in a wide range of applications.

## 4.4 COMPATIBILITY WITH DIFFERENT LLMS

Table 3: Compatibility of LLM-Exp with various LLMs. The human-norm scores (%) are recorded at the end of training and averaged across 3 random seeds. The underlined results indicate improvements over the raw RL algorithm, and the bold fonts indicate the best results.

| Environment | Double-DQN | Double-DQN+LLM-Exp | | | | |
|---|---|---|---|---|---|---|
| | | GPT-4o mini | GPT-4o | GPT-3.5 | Llama-3.1-405B | Llama-3.1-70B |
| Alien | 0.26 | 0.59 | 0.31 | 0.42 | **0.67** | 0.61 |
| Freeway | 17.75 | **69.71** | 67.27 | 66.45 | 60.22 | 63.7 |
| MsPacman | 1.56 | **2.75** | 1.63 | 1.53 | 1.88 | 2.01 |

In the framework work of LLM-Exp, we utilize the LLMs with text-only prompts, leveraging their text-processing capability to derive smart policy exploration strategies. Instead of relying on some specific types or versions of LLMs, our design is a general framework that can work with various types of LLMs. To evaluate this, besides GPT-4o mini used above, we test several other LLMs that

are most widely known, including GPT-4o[2], GPT-3.5[3], Llama-3.1-405B, and Llama-3.1-70B[4]. We train agents with the original Double-DQN algorithms, and then integrate Double-DQN with our LLM-Exp method, where the latter is driven by each of these different LLMs. In each experiment, we repeat the training process with three different random seeds and average the results. We summarize the game scores obtained at the end of training in Table 3 and show the learning curves in these experiments in Appendix A.2. In the results, our method consistently improves the human-normalized score of the original algorithms (14 out of 15 experiments) despite the type of LLMs, indicating its strong compatibility with different LLMs. We observe that GPT-4o mini tends to be the best choice for LLM-Exp, while the Llama model may outperform others in specific environments. It is also interesting to note that the performance of LLM-Exp is much worse when driven by GPT-4o than when driven by GPT-4o mini. The actual reason for this is worth future study, while one possible speculation is that the super LLMs, like GPT-4o, are too sophisticated, which tend to greedily fit specific actions instead of providing flexible policy exploration with randomness, thus limiting the performance.

## 4.5 PERFORMANCE VS COMPUTATIONAL CONSUMPTION

Table 4: Performance of LLM-Exp with various ablation designs. The human-norm scores (%) are recorded at the end of training and averaged across 3 random seeds. The underlined results indicate improvements over the raw RL algorithm, and the bold fonts indicate the best results.

| Environment | Double-DQN | Double-DQN+LLM-Exp | | | | | | | | |
|---|---|---|---|---|---|---|---|---|---|---|
| | | Full design | | | w/o summarize & suggestion | | | w/o environment information | | |
| | | Score | Token in (k) | Token out (k) | Score | Token in (k) | Token out (k) | Score | Token in (k) | Token out (k) |
| Alien | 0.26 | **0.59** | 248.73 | 179.59 | 0.51 | 111.07 | 112.54 | 0.38 | 186.41 | 165.90 |
| Freeway | 17.75 | **69.71** | 220.12 | 138.75 | 68.97 | 88.91 | 69.94 | 61.26 | 164.38 | 134.93 |
| MsPacman | 1.56 | **2.75** | 291.30 | 201.22 | 2.32 | 129.18 | 125.22 | 1.89 | 222.05 | 208.31 |

To facilitate the wide application of our method, it is important to understand the relationship between its performance and computational consumption. Since LLM-Exp is a simple plug-in design that does not impact the original computational consumption in RL training, we mainly focus on its auxiliary consumption in utilizing LLMs.

There exist two major trade-offs between the performance and computational cost of LLM-Exp, where the first one lies in the design of LLM workflow. To uncover the roles of several key components in the LLM workflow, we conduct ablation experiments. In one experiment, we remove the summarize & suggestion mechanism and allow a single LLM to directly output a probability distribution for future policy exploration based on the {*TaskDescription*}, {*ActionSequence*}, and {*EpisodeReward*}. In another experiment, we retain the two-stage design of the LLM workflow but do not provide the {*TaskDescription*}, only informing the LLMs of the environment's name. In each experiment, we repeat the training process with three different random seeds and average the results. As shown in Table 4 and Appendix A.1, both ablations continue to improve the performance of the original Double-DQN algorithm while significantly reducing the token consumption of LLM. However, the first ablation lacks sufficient analysis of the agent's learning status, making it less flexible for adjustment during the training process. The second ablation lacks sufficient environmental information, making it less adaptive to specific environments. As a result, neither of them performs as well as the full design of LLM-Exp.

The second trade-off lies in the setting of the two key parameters in our design, namely action sampling density ($M$) and exploration adjusting interval ($K$). By reducing sampling density, i.e., smaller $M$, or reducing the frequency of adjusting the exploration strategy, i.e., larger $K$, we can obviously reduce the token consumption of LLM. To evaluate the impact of these, we conduct experiments and show the results in Table 5 and Appendix A.1. As the results illustrate, LLM-Exp with either smaller $M$ or larger $K$ keeps improving the performance of the original Double-DQN algorithm. However, smaller $M$ provides insufficient information about the agent's real-time

---

[2] https://platform.openai.com/docs/models/gpt-4o
[3] https://platform.openai.com/docs/models/gpt-3-5-turbo
[4] https://ai.meta.com/blog/meta-llama-3-1

Table 5: Performance of LLM-Exp with different action sampling density $M$ and exploration adjusting interval $K$. The human-norm scores (%) are recorded at the end of training and averaged across 3 random seeds. The underlined results indicate improvements over the raw RL algorithm, and the bold fonts indicate the best results.

| Environment | Double-DQN | Double-DQN+LLM-Exp | | | |
|---|---|---|---|---|---|
| | | $M$=100,$K$=1 | $M$=50,$K$=1 | $M$=200,$K$=1 | $M$=100,$K$=2 |
| Alien | 0.26 | 0.59 | 0.51 | **0.83** | 0.38 |
| Freeway | 17.75 | **69.71** | 64.72 | 66.52 | 66.52 |
| MsPacman | 1.56 | **2.75** | 2.22 | 2.24 | 2.07 |

learning status and larger $K$ limits adjustments on the exploration strategy. As a result, both of them are less capable of flexibly adapting the policy exploration to the training process, achieving worse performance than LLM-Exp with the original settings of $M$ and $K$. Moreover, we also analyze the impact of increasing the sampling density, i.e., larger $M$. As the results indicate, although increasing the token consumption of LLM, a larger $M$ does not consistently improve the performance of LLM-Exp. This may be because the original settings of $M$ already provide sufficient information about the agent's real-time learning status. Therefore, further increasing the sampling density complicates the LLM's ability to analyze and summarize the data, which may hinder overall performance.

From these analyses, we demonstrate that the full design and properly configured values of $M$ and $K$ are critical for achieving the best performance of LLM-Exp. However, we also highlight the trade-offs between performance and computational consumption in LLM-Exp. Therefore, for deployments with limited computational resources, it is possible to simplify the design of the LLM workflow or adjust $M$ and $K$ as above to reduce computational consumption while still maintaining certain performance improvements over the original RL algorithm. For deployments with sufficient computational resources, the full design with the original settings of $M$ and $K$ is the optimal choice.

## 4.6 CASE STUDIES

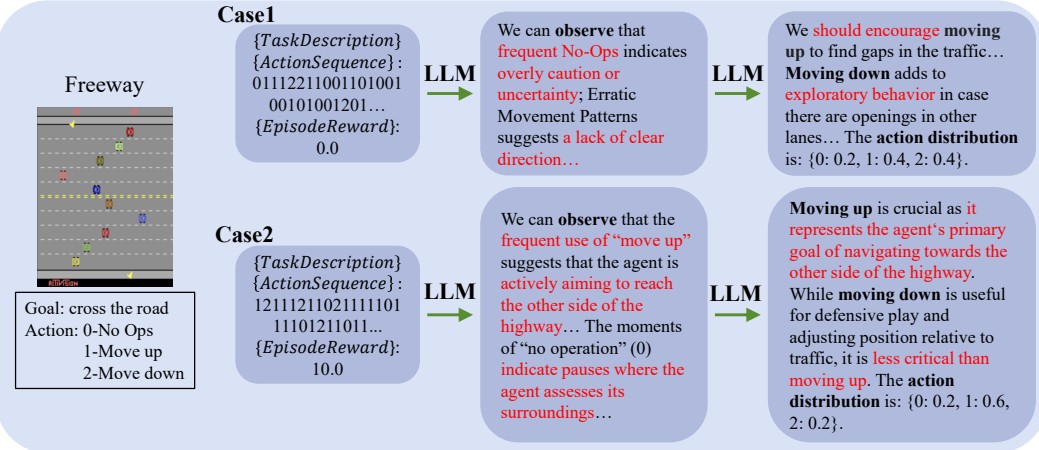

Figure 4: Case study of the operating process of LLM-Exp.

To demonstrate the rationality in determining the policy exploration strategy with LLM-Exp, we provide an intuitive case study in Figure 4 within the environment of the Freeway. In this environment, the goal is crossing the busy road safely, while the action space includes three items, namely no-ops, moving up, and moving down. In case 1, the previous action of the agent involves a large proportion of 'no ops', and the LLM in the stage of learning status summarizing points out its overly caution behavior that lacks clear direction. Subsequently, the latter LLM generates an exploration strategy that stresses moving up and down. In case 2, the previous action of the agent involves a large proportion of 'moving up', and the former LLM reveals that the current learning status of the agent

is actively aiming to reach the other side of the highway. Based on this, the latter LLM generates an exploration strategy that further encourages 'moving up' to reach the goal while also adding a small proportion of 'moving down' to adjust position relative to traffic for safety. Such rational analyses enable our design to generate smart policy exploration strategies that are adaptive to specific environments and learning processes, enhancing the performance of various RL algorithms.

## 5 Related Works

### 5.1 Policy Exploration in RL

Plentiful approaches have been proposed and are widely used in existing RL algorithms for policy exploration. One of the most basic methods is the $\epsilon$-greedy strategy used in DQN (Mnih et al., 2015), where with a probability of $\epsilon$, the agent randomly samples an action from all possible actions rather than greedily exploiting the current best one. As an improvement of DQN, Noisy-DQN introduces noisy networks (Fortunato et al., 2018), which inject randomness directly into the action selection process, allowing for better policy exploration. Other methods utilize the randomness introduced by Gaussian distributions. For example, the actions are sampled from Gaussian distributions in PPO (Schulman et al., 2017), and small Gaussian noises are added to the deterministic actions in DDPG (Lillicrap et al., 2016). Also, in some implementations of DDPG (Luck et al., 2019; Zhang et al., 2019; Yoo et al., 2021), the standard white Gaussian noise is replaced with an Ornstein-Uhlenbeck (OU) process with temporal correlation (Gillespie, 1996; Maller et al., 2009), leading to smoother and potentially more effective policy exploration. Moreover, extensive algorithms incorporate an entropy term in the reward function (Haarnoja et al., 2018; Zhao et al., 2019; Pitis et al., 2020), encouraging more diverse action selections to enhance policy exploration. However, these methods are designed based on prefixed stochastic processes, which can neither adapt to specific environments nor be flexibly adjusted during the training process. In contrast, we design to dynamically generate a stochastic process by LLMs to guide policy exploration, which is adaptive and flexible.

### 5.2 Enhancing RL with LLMs

As two important directions in current AI research, many studies have explored the use of LLMs in enhancing the performance of RL (Cao et al., 2024). First, a significant body of work focuses on leveraging LLMs to design reward functions based on the characteristics of the environment and tasks, providing feedback for the agent's policy learning (Colas et al., 2023; Wu et al., 2024; Song et al., 2023; Xie et al., 2024). Additionally, other research explores using LLMs to design state representation functions, offering more effective state inputs for the agents (Wang et al., 2024a). On a macro level, LLMs have been utilized to decompose complex tasks into sub-goals (Colas et al., 2023) or provide high-level instructions (Zhou et al., 2023) to facilitate RL training. Moreover, LLMs are employed in human-AI coordination, enabling humans to specify the desired strategies for RL agents through natural language instructions (Hu & Sadigh, 2023). Despite these works, it remains largely unexplored how to leverage LLMs to enhance policy exploration in RL remains largely unexplored, and the paper conducts investigations to bridge such knowledge gap.

## 6 Conclusions

In this paper, we propose to improve the policy exploration in RL with LLMs. We design to use LLMs to analyze the agent's real-time learning status based on its action-reward trajectory and then periodically update the probability distribution for policy exploration. By doing so, we are able to adapt the policy exploration to any specific environment and flexibly adjust it during the training process, only with the requirement of low-cost text-only prompts. Through extensive experiments and in-depth analyses in various environments, we verify the validity of our design and illustrate its compatibility with a wide range of established RL algorithms. One direction worth future studies lies in further combining our method with RL algorithms in continuous action space. It may be feasible to prompt the LLMs to generate some offset and stretch parameters and thus flexibly shape the Gaussian distribution used for policy exploration in algorithms like DDPG and PPO according to the environmental characteristics and the real-time learning status of the agent.

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

# A APPENDIX

## A.1 IMPLEMENTATION DETAILS

In this section, we provide all implementation details for reproducibility in Table 6. Please refer to our source code at `https://anonymous.4open.science/r/LLM-Exp-4658` for more details.

Table 6: Implementation details

| Module | Element | Detail |
|---|---|---|
| System | OS | Ubuntu 22.04.2 |
| | CUDA | 11.7 |
| | Python | 3.11.4 |
| | Device | 8*NVIDIA A100 80G |
| Double-DQN | Gamma | 0.99 |
| | Batch Size | 256 |
| | Interval of target network updating | 1000 |
| | Optimizer | Adam |
| | Learning rate | 0.0001 |
| | Replay buffer size | 10000 |
| | Start epsilon | 1 |
| | Min epsilon | 0.1 |
| | Epsilon decay per step | 0.99999 |
| Learning status summarizing | Model name | gpt-4o-mini-2024-07-18 |
| | Temperature | 1.0 |
| Policy exploration strategy generation | Model name | gpt-4o-mini-2024-07-18 |
| | Temperature | 1.0 |
| Test of different LLMs | Model name for GPT-4o | gpt-4o-2024-08-06 |
| | Temperature for GPT-4o | 1.0 |
| | Model name for GPT-3.5 | gpt-3.5-turbo-0125 |
| | Temperature for GPT-3.5 | 1.0 |
| | Model name for Llama-3.1-405B | Llama-3.1-405B-Instruct |
| | Temperature for Llama-3.1-405B | 1.0 |
| | Model name for Llama-3.1-70B | Llama-3.1-70B-Instruct |
| | Temperature for Llama-3.1-70B | 1.0 |

### A.2 SUPPLEMENTARY RESULTS

Here, we show the learning curves in the experiments of the main texts.

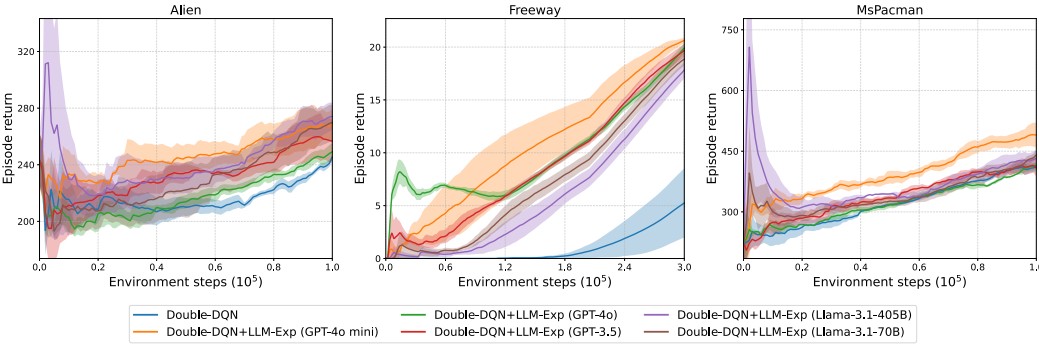

Figure 5: Compatibility of LLM-Exp with various LLMs. In each experiment, we repeatedly run the training process with three different random seeds and use the shaded area to indicate the standard deviations.

In Figure 5, we show the training process with the original Double-DQN algorithms, and then integrating Double-DQN with our LLM-Exp method, where the latter is driven different LLMs. In the results, our method consistently improves the human-normalized score of the original algorithms (14 out of 15 experiments) despite the type of LLMs, indicating its strong compatibility with different LLMs.

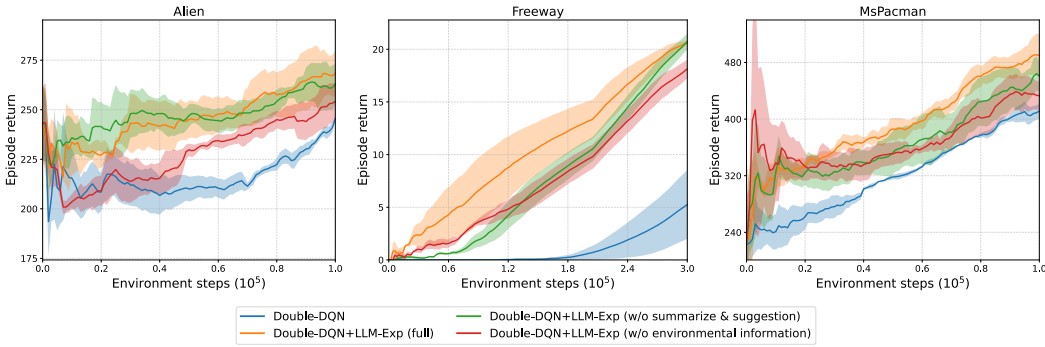

Figure 6: Performance of LLM-Exp with various ablation designs. In each experiment, we repeatedly run the training process with three different random seeds and use the shaded area to indicate the standard deviations.

In Figure 6, we show the training process with the original Double-DQN algorithms, and then integrating Double-DQN with our LLM-Exp method, where the latter contains different ablation designs. In the results, both ablations continue to improve the performance of the original Double-DQN algorithm while significantly reducing the token consumption of LLM. However, the first ablation lacks sufficient analysis of the agent's learning status, making it less flexible for adjustment during the training process. The second ablation lacks sufficient environmental information, making it less adaptive to specific environments. As a result, neither of them performs as well as the full design of LLM-Exp.

In Figure 7, we show the training process with the original Double-DQN algorithms, and then integrating Double-DQN with our LLM-Exp method, where the latter is configured with different values of $M$. In the results, LLM-Exp with smaller $M$ keeps improving the performance of the original Double-DQN algorithm. However, smaller $M$ provides insufficient information about the agent's real-time learning status, achieving worse performance than LLM-Exp with the original settings of $M$.

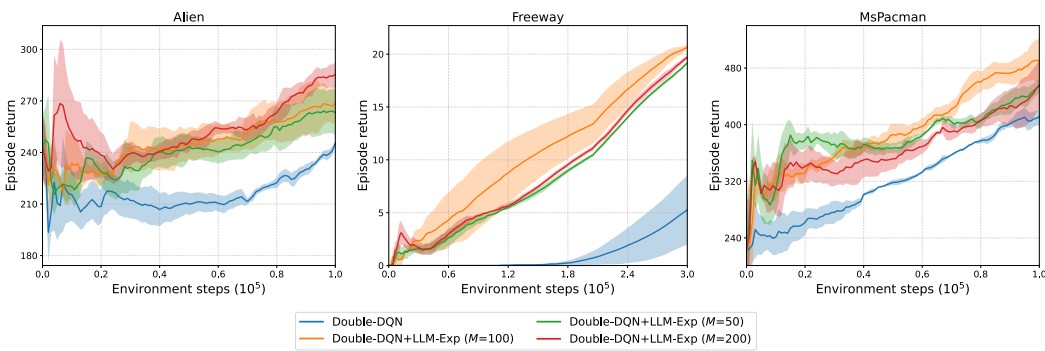

Figure 7: Performance of LLM-Exp with different action sampling density $M$. In each experiment, we repeatedly run the training process with three different random seeds and use the shaded area to indicate the standard deviations.

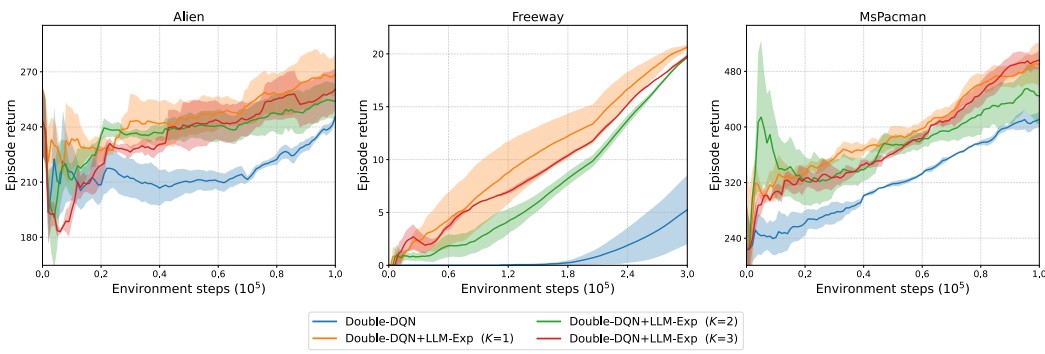

Figure 8: Performance of LLM-Exp with different exploration adjusting interval $K$. In each experiment, we repeatedly run the training process with three different random seeds and use the shaded area to indicate the standard deviations.

In Figure 8, we show the training process with the original Double-DQN algorithms, and then integrating Double-DQN with our LLM-Exp method, where the latter is configured with different values of $K$. In the results, LLM-Exp with larger $K$ keeps improving the performance of the original Double-DQN algorithm. However, larger $K$ limits adjustments on the exploration strategy, achieving worse performance than LLM-Exp with the original settings of $K$.

In all the figures above, we repeat the training process with three different random seeds in each experiment and average the results. We use the shaded area to indicate the standard deviations.

### A.3 DETAILED PROMPTS

Here we list the detailed {*TaskDescription*} in the prompts for Atari environments.

- **Alien**: The task is a reinforcement learning problem where an agent controls an astronaut navigating through a dangerous alien world. The action space is discrete with 18 options: {0: no operation, 1: fire, 2: move up, 3: move right, 4: move left, 5: move down, 6: move up-right, 7: move up-left, 8: move down-right, 9: move down-left, 10: move up and fire, 11: move right and fire, 12: move left and fire, 13: move down and fire, 14: move up-right and fire, 15: move up-left and fire, 16: move down-right and fire, 17: move down-left and fire}. In the environment, the agent receives +50 points for defeating an alien and +100 points for clearing a level. Small rewards like +10 points are given for collecting power-ups, while penalties include -50 points for taking damage and -100 points for losing a life. The game ends when the agent loses all lives, with the goal being to maximize cumulative rewards through effective combat, exploration, and survival.

- **Amidar**: The task is a reinforcement learning problem where an agent controls a character navigating a maze to avoid enemies and complete objectives by marking sections of the maze. The action space is discrete with 10 options: {0: no operation, 1: fire, 2: move up, 3: move right, 4: move left, 5: move down, 6: move up and fire, 7: move right and fire, 8: move left and fire, 9: move down and fire}. In the environment, the fire action has no functional effect, as the primary objective is to move through the maze. The observation space consists of raw pixel values representing the game screen, showing the character, enemies, and the maze layout. The agent receives +10 points for marking a section of the maze and +50 points for completing an entire maze level. Additionally, the agent earns +100 points for capturing an enemy while in a powered-up state, and +20 points for collecting special bonus items scattered throughout the environment. However, the agent is penalized with -50 points for being caught by an enemy, and an additional -5 points for excessive inaction or idling for too long. The game ends when the agent loses all lives or completes the entire maze. The goal is to maximize the score by navigating the maze efficiently while avoiding enemies.

- **BankHeist**: The task is a reinforcement learning problem where an agent controls a character involved in a bank heist, navigating through a dynamic environment filled with guards and obstacles. The action space is discrete with 18 options: {0: no operation, 1: fire, 2: move up, 3: move right, 4: move left, 5: move down, 6: move up-right, 7: move up-left, 8: move down-right, 9: move down-left, 10: move up and fire, 11: move right and fire, 12: move left and fire, 13: move down and fire, 14: move up-right and fire, 15: move up-left and fire, 16: move down-right and fire, 17: move down-left and fire}. The observation space consists of raw pixel values representing the game screen, showing the agent, guards, and loot. In this environment, the agent receives rewards for successfully stealing loot and evading or neutralizing guards. The game ends when the agent loses all lives, and the primary objective is to maximize cumulative rewards through stealthy navigation, effective shooting, and strategic interactions with the environment.

- **Breakout**: The task is a reinforcement learning problem where an agent controls a paddle at the bottom of the screen, aiming to hit a ball and break bricks at the top. The action space is discrete with 4 options: {0: no operation, 1: fire (launch the ball), 2: move right, 3: move left}. The observation space consists of raw pixel values representing the game screen, displaying the paddle, the ball, and the bricks. The reward mechanism is designed to incentivize the destruction of bricks, with the agent earning points each time a brick is broken. In this reward mechanism, players score points by hitting bricks of various colors with a ball. Each brick color is assigned a specific point value: red and orange bricks yield 7 points, yellow and green bricks grant 4 points, while aqua and blue bricks provide 1 point each. The game ends when the agent loses all its lives by failing to catch the ball with the paddle. The primary objective is to maximize cumulative rewards by strategically controlling the paddle to keep the ball in play and target higher-value bricks while avoiding misses.

- **ChopperCommand**: The task is a reinforcement learning problem where an agent controls a helicopter navigating through a desert environment filled with enemy vehicles and aircraft. The action space is discrete with 18 options: {0: no operation, 1: fire, 2: move

up, 3: move right, 4: move left, 5: move down, 6: move up-right, 7: move up-left, 8: move down-right, 9: move down-left, 10: move up and fire, 11: move right and fire, 12: move left and fire, 13: move down and fire, 14: move up-right and fire, 15: move up-left and fire, 16: move down-right and fire, 17: move down-left and fire}. The observation space consists of raw pixel values representing the game screen, displaying the helicopter, enemy vehicles, aircraft, and fuel depots. In this reward design mechanism, players earn points by shooting down enemy aircraft: 100 points for each enemy helicopter and 200 points for each enemy jet. A bonus is awarded for destroying an entire wave of hostile aircraft, calculated by multiplying the number of remaining trucks in the convoy by the wave number (from one to ten) and then by 100. This system incentivizes players to maximize their score through both individual kills and strategic gameplay. The game ends when the agent runs out of fuel or is hit by enemy fire and loses all lives. The primary objective is to maximize cumulative rewards by skillfully navigating the environment, destroying enemies, collecting fuel, and avoiding hazards to survive as long as possible.

- **CrazyClimber**: The task is a reinforcement learning problem where an agent controls a climber scaling the side of a tall building while avoiding various obstacles. The action space is discrete with 9 options: {0: no operation, 1: move up, 2: move right, 3: move left, 4: move down, 5: move up-right, 6: move up-left, 7: move down-right, 8: move down-left}. The observation space consists of raw pixel values representing the game screen, displaying the climber, the building, windows, and various obstacles such as falling objects. In the reward mechanism, players earn points in two ways: climbing points for each row of windows climbed and bonus points for reaching the top of each skyscraper. The climbing points vary by building, with 100 points per row for Building 1, 200 for Building 2, 300 for Building 3, and 400 for Building 4. Bonus points serve as a timer; they start at a maximum value when climbing a new building and decrease by 100 points every ten seconds. To retain bonus points, players must reach the top and grab the helicopter within 30 seconds, as bonus points continue to decline until the helicopter is reached. The maximum bonus points also increase with each building, ranging from 100,000 points for Building 1 to 400,000 points for Building 4. The game ends when the climber falls or loses all lives. The primary objective is to maximize cumulative rewards by skillfully navigating the vertical environment, dodging hazards, and climbing as high as possible without falling.

- **Freeway**: The task is a reinforcement learning problem where an agent controls a character attempting to cross a busy highway filled with fast-moving cars. The action space is discrete with 3 options: {0: no operation, 1: move up, 2: move down}. The observation space consists of raw pixel values representing the game screen, displaying the character, various lanes of traffic, and the road. The reward mechanism is designed to incentivize the successful crossing of the highway. The agent earns points for reaching the other side of the road, with each successful crossing awarding a fixed number of points. There are no explicit negative rewards, but the agent loses time and progress when hit by a car, as it is sent back to the starting point. The game ends when a time limit is reached. The primary objective is to maximize cumulative rewards by skillfully navigating through the traffic, avoiding cars, and making as many successful crossings as possible before time runs out.

- **Hero**: The task is a reinforcement learning problem where an agent controls a hero navigating through an underground cave system filled with enemies and obstacles. The action space is discrete with 18 options: {0: no operation, 1: fire, 2: move up, 3: move right, 4: move left, 5: move down, 6: move up-right, 7: move up-left, 8: move down-right, 9: move down-left, 10: move up and fire, 11: move right and fire, 12: move left and fire, 13: move down and fire, 14: move up-right and fire, 15: move up-left and fire, 16: move down-right and fire, 17: move down-left and fire}. The observation space consists of raw pixel values representing the game screen, showing the hero, enemies, environmental hazards, and collectible items. The reward mechanism is designed to incentivize the exploration of the cave and the collection of various items, such as treasure. The agent earns points for defeating enemies and gathering treasures scattered throughout the cave. The hero may also gain points by rescuing trapped miners. There are penalties for losing health due to enemy attacks or environmental hazards. The game ends when all lives are lost. The primary objective is to maximize cumulative rewards by skillfully navigating the cave system, defeating enemies, avoiding hazards, and collecting valuable items.

- **Jamesbond**: The task is a reinforcement learning problem where an agent controls James Bond navigating through various action-packed levels filled with enemies and obstacles. The action space is discrete with 18 options: {0: no operation, 1: fire, 2: move up, 3: move right, 4: move left, 5: move down, 6: move up-right, 7: move up-left, 8: move down-right, 9: move down-left, 10: move up and fire, 11: move right and fire, 12: move left and fire, 13: move down and fire, 14: move up-right and fire, 15: move up-left and fire, 16: move down-right and fire, 17: move down-left and fire}. The observation space consists of raw pixel values representing the game screen, displaying James Bond, various enemies, vehicles, and obstacles. In this reward system, players earn points by collecting various targets. For the reward system, each target has the following point value: a Diamond is worth 50 points, while the Frogman, Space Shuttle, and Submarine each provide 200 points. The Poison Bomb and Torpedo are worth 100 points each. The Spinning Satellite offers the highest reward at 500 points, while the Rapid Rocket and Fire Bomb also contribute 100 points each. Completing the mission yields a substantial bonus of 5,000 points. This design encourages players to explore actively and prioritize collecting high-value targets to maximize their cumulative score. The game ends when all lives are lost. The primary objective is to maximize cumulative rewards by skillfully navigating the levels, shooting enemies, and strategically completing missions while avoiding hazards and enemy attacks.

- **Krull**: The task is a reinforcement learning problem where an agent controls a character navigating through a vibrant fantasy world filled with enemies, moving platforms, and obstacles. The action space is discrete with 18 options: {0: no operation, 1: fire, 2: move up, 3: move right, 4: move left, 5: move down, 6: move up-right, 7: move up-left, 8: move down-right, 9: move down-left, 10: move up and fire, 11: move right and fire, 12: move left and fire, 13: move down and fire, 14: move up-right and fire, 15: move up-left and fire, 16: move down-right and fire, 17: move down-left and fire}. The observation space consists of raw pixel values representing the game screen, displaying the character, various enemies, laser barriers, and collectible items such as gems and keys. The reward mechanism is designed to incentivize progressing through different rooms by collecting keys to unlock doors and defeating enemies with laser shots. The agent earns points for defeating enemies, collecting gems, and clearing levels. The game becomes progressively more difficult with more enemies and complex rooms to navigate. The game ends when all lives are lost or when the player completes all levels. The primary objective is to maximize cumulative rewards by skillfully navigating the environment, defeating enemies, avoiding hazards, and collecting items to progress through the world.

- **MsPacman**: The task is a reinforcement learning problem where an agent controls Ms. Pacman navigating through a maze filled with pellets, power-ups, and enemy ghosts. The action space is discrete with 9 options: {0: no operation, 1: move up, 2: move right, 3: move left, 4: move down, 5: move up-right, 6: move up-left, 7: move down-right, 8: move down-left}. The observation space consists of raw pixel values representing the game screen, displaying Ms. Pacman, pellets, power pellets, and ghosts moving around the maze. The reward mechanism is designed to incentivize the collection of pellets and the strategic use of power-ups. Ms. Pacman earns points for each pellet collected and additional points for eating ghosts after consuming a power pellet. However, if she gets caught by a ghost without the power-up, a life is lost. The game ends when all lives are lost or when all pellets in the maze are collected. The primary objective is to maximize cumulative rewards by skillfully navigating the maze, avoiding or chasing ghosts when appropriate, and collecting as many pellets and power-ups as possible.

- **Pong**: The task is a reinforcement learning problem where an agent controls a paddle to hit a ball and score points by getting the ball past the opponent's paddle. The action space is discrete with 6 options: {0: no operation, 1: fire, 2: move the paddle up, 3: move the paddle down, 4: right fire, 5: left fire}. In the environment, the fire action has no functional effect, as we can only move the paddle up and down. The observation space consists of raw pixel values representing the game screen. The agent receives a reward of +1 for scoring and -1 when the opponent scores. The game ends when either side reaches 21 points.

- **Qbert**: The task is a reinforcement learning problem where an agent controls Qbert, a character navigating through a pyramid of cubes while avoiding enemies and hazards. The action space is discrete with 6 options: {0: no operation, 1: fire (jump), 2: move up, 3: move right, 4: move left, 5: move down}. The observation space consists of raw pixel

values representing the game screen, displaying Qbert, enemies, and the pyramid of cubes that Qbert must jump on to change their color. The reward mechanism is designed to incentivize jumping on cubes and avoiding enemies. Qbert earns points for each successful jump that changes the color of a cube, and additional points for completing a level by changing all cubes to the desired color. Penalties occur if Qbert is hit by enemies or falls off the pyramid, resulting in a lost life. The game ends when all lives are lost. The primary objective is to maximize cumulative rewards by skillfully navigating the pyramid, changing the colors of cubes, avoiding enemies, and completing levels efficiently.

- **Seaquest**: The task is a reinforcement learning problem where an agent controls a submarine navigating through an underwater world filled with enemy submarines, divers, and obstacles. The action space is discrete with 18 options: {0: no operation, 1: fire, 2: move up, 3: move right, 4: move left, 5: move down, 6: move up-right, 7: move up-left, 8: move down-right, 9: move down-left, 10: move up and fire, 11: move right and fire, 12: move left and fire, 13: move down and fire, 14: move up-right and fire, 15: move up-left and fire, 16: move down-right and fire, 17: move down-left and fire}. The observation space consists of raw pixel values representing the game screen, displaying the submarine, enemies, friendly divers, and the underwater environment. The reward mechanism is designed to incentivize the destruction of enemy submarines and the rescue of divers. The agent earns points for shooting enemy submarines and other hostile underwater threats, as well as for rescuing divers and bringing them safely to the surface. Penalties occur if the submarine is hit by enemy fire or runs out of oxygen, which results in a loss of life. The game ends when all lives are lost. The primary objective is to maximize cumulative rewards by skillfully navigating the underwater environment, avoiding enemies, rescuing divers, and managing oxygen levels effectively.

- **UpNDown**: The task is a reinforcement learning problem where an agent controls a car navigating through a colorful, fast-paced world filled with other vehicles and obstacles on winding roads. The action space is discrete with 6 options: {0: no operation, 1: fire, 2: move up, 3: move down, 4: move up and fire, 5: move down and fire}. The observation space consists of raw pixel values representing the game screen, displaying the agent's car, other vehicles, and road obstacles. The reward mechanism is designed to incentivize avoiding collisions and overtaking other vehicles. The agent earns points for passing other cars on the road and avoiding crashes. Higher rewards are earned by overtaking more cars and successfully navigating tricky sections of the road. The game ends when the agent collides with another car or falls off the road, resulting in a loss of life. The primary objective is to maximize cumulative rewards by skillfully maneuvering the car, avoiding collisions, overtaking as many vehicles as possible, and progressing through the levels without losing lives.

