# OpenReview forum: "LLM-Exp: Exploring the Policy in Reinforcement Learning with Large Language Models"
_ICLR.cc/2025/Conference — Submitted to ICLR 2025_

### Official Review · Reviewer_pE3k · 2024-10-17

**Soundness:** 2
**Presentation:** 2
**Contribution:** 2
**Rating:** 3
**Confidence:** 4

**Summary:**

LLM-Exp proposes the use of large language models (LLMs) to adjust the exploration distribution in deep Q-learning. The method uses one LLM to summarize action and reward trajectories as text, and another LLM to suggest an action distribution given that summary. The distribution is updated every K episodes. The method is evaluated with several LLMs, several DQN methods, and evaluated over Atari games.

**Strengths:**

The strengths of this paper are clarity and ease of method implementation. The method is both well described and easy to comprehend.

**Weaknesses:**

The main weaknesses of this paper are evaluation and magnitude of the improvement:
1) No measure of spread is given in the tables, which makes is very difficult to determine how significant the results are.
2) The improvement seems small on most environments other than Pacman, Freeway, Alien, and Hero.
3) Because a task description is given, it is hard to know if the LLM has memorized what works well in these environments; will this method work on environments that are described but not with well known solutions, like Pacman?
4) This method works well on simple environments where the action distribution is low dimensional and has a similar shape among many states, but I am fairly skeptical that it will transfer to domains with high-dimensional action spaces (e.g. a control domain such as MuJoCo) or domains where the exploration required is much more complicated and there is not a single distribution that works well across states.
5) There are no comparisons to other common exploration methods, such as Random Network Distillation.
6) There are no comparisons to other methods using LLMS, e.g. CLIP embeddings (e.g. Tam et al., 2022)
7) This evaluation seems fairly limited in that it is tested with DQN (and only works for small discrete action spaces)

**Questions:**

1) Can you report the standard deviation in the tables? (Or some measure of spread.)
2) The ablation with only 1 LLM seems to save a very large amount of compute while barely affecting the performance; am I missing something?

---

### Official Review · Reviewer_CR9n · 2024-10-30

**Soundness:** 1
**Presentation:** 2
**Contribution:** 1
**Rating:** 3
**Confidence:** 3

**Summary:**

The paper proposes LLM-Exp, a novel approach to enhancing policy exploration in reinforcement learning (RL) using large language models (LLMs). LLM-Exp aims to adaptively adjust the exploration strategy of RL agents by leveraging LLMs to analyze the agent's learning status and generate a probability distribution for exploration. The method integrates with DQN-based algorithms and has been tested on Atari benchmarks.

**Strengths:**

1. Exploration is a hard and important problem in RL.

2. The paper provides a clear explanation of LLM-Exp’s framework and the role of each LLM.

**Weaknesses:**

Action Space Limitation: A significant limitation is that LLM-Exp is restricted to discrete action spaces, demonstrated solely on Atari environments. Expanding this method to handle continuous action spaces—common in many real-world applications like robotics—is crucial for broader applicability.

Insufficient Baselines: The paper lacks a comparison with other methods that use LLMs directly as value functions or policies and only compared to Double DQN's variants.

Related Work and Novelty: The current review of related work does not thoroughly address recent advances that apply LLMs directly within RL frameworks.

Experimental Rigor: Experiments were run using three random seeds, fewer than the typical five or more used in Atari studies to ensure robustness.

**Questions:**

Could the authors elaborate on potential adaptations of LLM-Exp for continuous action spaces? Given the importance of continuous control tasks, particularly in robotics, this seems a key direction to expand LLM-Exp’s applicability.

Given the sensitivity of LLM-Exp to sampling parameters (M, K), what strategies would the authors recommend for optimizing these parameters in different environments or tasks?

---

### Official Review · Reviewer_VmYj · 2024-10-31

**Soundness:** 2
**Presentation:** 3
**Contribution:** 2
**Rating:** 3
**Confidence:** 3

**Summary:**

This paper proposes LLM-Exp to improve the policy exploration in RL training with large language models (LLMs). Specifically, they use an LLM to adapt the exploration scheme with the training tasks and training progress. This improvement can be broadly integrated with various deep Q-learning algorithms.

**Strengths:**

This paper targets at an important problem in RL, that is, the exploration-exploitation tradeoff. The idea to adapt the exploration scheme based on a LLM is interesting and novel. The authors also provide a diverse range of evaluation results.

**Weaknesses:**

(a) The technical contributions appear somewhat limited, which is covered in Figure 1.

(b) The performance of DDQN and DDQN+LLM-Exp on Atari falls significantly short of state-of-the-art (SOTA) benchmarks. The original DDQN paper ("Deep Reinforcement Learning with Double Q-learning," 2015, page 10) reports much stronger results than those presented in Table 1, not to mention the performance achieved in more recent Atari studies. Demonstrating that LLM-Exp can drive performance beyond SOTA would be critical.

**Questions:**

Please see the weakness part.

---

### Official Review · Reviewer_H1tJ · 2024-11-04

**Soundness:** 3
**Presentation:** 3
**Contribution:** 2
**Rating:** 5
**Confidence:** 4

**Summary:**

The paper mainly utilizes two off-the-shelf LLMs to boost the exploration of RL algorithms, with experiments conducted on the Atari benchmark. One LLM is used to analyze the policy status from a recent action-reward trajectory, and another is used to generate the policy exploration strategy according to the analyzed status.

**Strengths:**

- The idea of using LLMs to improve policy exploration is interesting.
- The proposed method is clearly presented.
- The proposed method is implemented by various backbones in experiments.

**Weaknesses:**

- The way of using LLMs to improve the exploration strategies is kind of trivial. LLMs contain a broad spectrum of world knowledge, which could be more useful if used for developing generalist RL agents, e.g., for acting as the decision-making agents, for zero-shot generalization, for developing foundation RL models.
- The necessity of using LLMs for exploration is not well demonstrated and verified. Exploration is one of the oldest problems in RL, and many studies have been investigated from various perspectives. The advantages over existing works involving exploration (especially those lightweight designs) are not elaborated convincingly and are not demonstrated by experiments.
- The proposed method only feeds the action-reward trajectory into the LLM for policy status analysis, which is counter-intuitive. For an artificially intelligent agent, it will analyze its current status from the state that describes its situation in the world. Also, excluding the state from the status analysis seems to lighten the burden of the employed LLM, but will largely limit the performance upper bound of the proposed method.
- The core pipeline of the proposed method is the design of the two prompts, which is specialized according to the Atari domain. Its universality to general domains remains unknown.

**Questions:**

- Why using two LLMs, since one LLM can do both works of status analysis and strategy recommendation?

---

### Meta-Review · Area_Chair_SkDm · 2024-12-08

**Metareview:**

The paper leverages LLM for exploration in RL training with large language models (LLMs). All reviews are negative and raise many weaknesses including:
- limited to discrete action space;
- technical contributions are limited;
- empirical experiments are insignificant in terms of baselines and improvement;

As the authors did not provide a rebuttal, thus it is a clear rejection.

**Additional Comments On Reviewer Discussion:**

The authors did not provide a rebuttal.

---

### Decision · Program_Chairs · 2025-01-22

Reject